# Combined inhibition of KRAS[G12C] and mTORC1 kinase is synergistic in non-small cell lung cancer

Hidenori Kitai [1,7], Philip H. Choi[1,7], Yu C. Yang [2], Jacob A. Boyer[1], Adele Whaley[1], Priya Pancholi [1], Claire Thant[1], Jason Reiter[1], Kevin Chen[3], Vladimir Markov[3], Hirokazu Taniguchi[4], Rui Yamaguchi [5], Hiromichi Ebi [6], James Evans[2], Jingjing Jiang[2], Bianca Lee[2], David Wildes [2], Elisa de Stanchina[3], Jacqueline A. M. Smith[2], Mallika Singh [2] ✉ & Neal Rosen [1] ✉

Current KRAS[G12C] (OFF) inhibitors that target inactive GDP-bound KRAS[G12C] cause responses in less than half of patients and these responses are not durable. A class of RAS[G12C] (ON) inhibitors that targets active GTP-bound KRAS[G12C] blocks ERK signaling more potently than the inactive-state inhibitors. Sensitivity to either class of agents is strongly correlated with inhibition of mTORC1 activity. We have previously shown that PI3K/mTOR and ERK-signaling pathways converge on key cellular processes and that inhibition of both pathways is required for inhibition of these processes and for significant antitumor activity. We find here that the combination of a KRAS[G12C] inhibitor with a selective mTORC1 kinase inhibitor causes synergistic inhibition of Cyclin D1 expression and cap-dependent translation. Moreover, BIM upregulation by KRAS[G12C] inhibition and inhibition of MCL-1 expression by the mTORC1 inhibitor are both required to induce significant cell death. In vivo, this combination causes deep, durable tumor regressions and is well tolerated. This study suggests that the ERK and PI3K/mTOR pathways each mitigate the effects of inhibition of the other and that combinatorial inhibition is a potential strategy for treating KRAS[G12C]-dependent lung cancer.

Oncogenic mutants of KRAS are detected in 20–30% of non-small cell lung cancers (NSCLC), of which *KRAS[G12C]* is the most common mutation variant[1,2]. KRAS is a GTPase which cycles between an inactive [guanosine diphosphate (GDP)-bound] state and an active [guanosine triphosphate (GTP)-bound] state. Recently, small molecules have been developed that bind covalently to cysteine 12 within the switch II pocket of GDP-bound KRAS[G12C] [3–5] which blocks the binding of the guanine-nucleotide exchange factor (GEF) SOS, resulting in reduced levels of activated GTP-bound KRAS[G12C] [6]. RAS activity falls slowly, as a function of KRAS[G12C] GTPase activity[6]. In clinical trials, the inactive-state KRAS[G12C] mutant-specific inhibitors adagrasib (MRTX849) and sotorasib (AMG510) achieved 37–43% anti-tumor response rates in NSCLC patients with most experiencing disease progression 6–12 months after initiation of therapy[7–9]. Multiple mechanisms of acquired resistance have been identified, which largely converge on re-activation of the RAS pathway or alternatively activate parallel

[1]Program in Molecular Pharmacology and Department of Medicine, Memorial Sloan Kettering Cancer Center, New York, NY, USA. [2]Department of Biology, Revolution Medicines Inc., Redwood City, CA, USA. [3]Antitumor Assessment Core, Memorial Sloan Kettering Cancer Center, New York, NY, USA. [4]Department of Medicine, Thoracic Oncology Service, Memorial Sloan Kettering Cancer Center, New York, NY, USA. [5]Division of Cancer Systems Biology, Aichi Cancer Center Research Institute, Nagoya, Aichi 464-8681, Japan. [6]Division of Molecular Therapeutics, Aichi Cancer Center Research Institute, Nagoya, Aichi 464-8681, Japan. [7]These authors contributed equally: Hidenori Kitai, Philip H. Choi. ✉e-mail: msingh@revmed.com; rosenn@mskcc.org

signaling pathways such as the PI3K/mTOR pathway[10–14]. Improved and durable efficacy will require more potent inhibitors, and/or combination therapies that address resistance to the KRAS[G12C] inhibitor or that synergize with KRAS[G12C] inhibition to slow tumor growth and enhance cell death. We have previously shown that combined inhibition of MEK and AKT attenuates the growth of KRAS tumors and that this is dependent on suppression of 4E-BP1 phosphorylation[15].

Here, we show that KRAS[G12C] inhibition is associated with potent inhibition of ERK signaling but less potent and variable inhibition of PI3K/mTOR signaling, and that the sensitivity of tumor cell lines to KRAS[G12C] inhibition is closely correlated with their ability to inhibit mTOR substrate phosphorylation. Combining a KRAS[G12C] inhibitor with inhibitors of PI3K, AKT, or mTOR causes synergistic inhibition of cell growth and induction of apoptosis in vitro, with mTORC1-selective kinase inhibitors being most active. In xenograft models of mutant KRAS[G12C]-dependent NSCLC in vivo, combining a KRAS[G12C] inhibitor with a mTORC1-selective kinase inhibitor causes synergistic durable and deep tumor regression at tolerated doses. We find that the mechanistic basis for the observed combinatorial antitumor activity is dependent on the concomitant inhibition of both pathways which converges on several key processes, including Cyclin D1 expression, cap-dependent translation, and the anti-apoptotic network, leading to synergistic inhibition of proliferation and induction of apoptosis.

## Results

### Active-state KRAS[G12C] inhibitors inhibit ERK pathway output more potently and durably than inactive-state KRAS[G12C] inhibitors

The effects of pharmacologic inhibition of KRAS[G12C] in NSCLC lines were first studied with three inactive-state KRAS[G12C] inhibitors, sotorasib[16], adagrasib[17], and AZD8037[18]. After 4 h of incubation, AZD8037 inhibited ERK phosphorylation (pERK) at concentrations between 100 nM and 1000 nM in the KRAS[G12C] NSCLC cell lines H358 and H23 but did not affect pERK in the KRAS[G12S] A549 lung cancer cell line (Supplementary Fig. 1a). Cell viability was assessed after incubation of 12 KRAS[G12C] NSCLC lines with varying concentrations of sotorasib, adagrasib, or AZD8037 for 72 h (Fig. 1a). At 1 μM, sensitivity of cell lines to KRAS[G12C] inhibition was variable, but each cell line responded similarly to all three compounds.

We asked whether these observed differences in sensitivity reflect differences in potency or duration of inhibition of RAS signaling. To this end, we utilized a class of active-state RAS[G12C] inhibitors which includes the preclinical tool compounds RM-018 and RMC-4998, the latter being more potent and exhibiting oral bioavailability in vivo[13,19]. These preclinical tool compounds are representative of the investigational agent RMC-6291 which is currently in early-phase clinical trials. These compounds form a high-affinity tri-complex with cyclophilin A and GTP-bound KRAS[G12C], followed by covalent bond formation, blocking the association with downstream effectors. The rate of inhibition of KRAS[G12C] by inactive-state KRAS[G12C] inhibitors is determined by the GTPase activity of the mutant. By contrast, RM-018 and RMC-4998 directly bind to GTP-bound KRAS[G12C] which results in more rapid and sustained inhibition[19]. RM-018 inhibited ERK and PI3K/AKT/mTOR signaling in the KRAS[G12C] cell lines LU65 and H23 in a concentration-dependent manner and caused a mobility shift in KRAS that indicated covalent binding (Supplementary Fig. 1b). By contrast, RM-018 had little or no effect on ERK inhibition or the cell growth of A549, a KRAS[G12S] mutant NSCLC cell line or in PC-9, a NSCLC cell line with an EGFR mutation and KRAS[WT], consistent with the specificity of this inhibitor for KRAS[G12C] (Supplementary Fig. 1c, d).

The effects of RM-018 and RMC-4998 were further evaluated in the panel of KRAS[G12C] cell lines. H358, LU65, and H2122 cells were the most sensitive to RM-018, with IC50 values more than 10-fold lower than those of adagrasib (Fig. 1b). However, those cell lines most resistant to the inactive-state KRAS[G12C] inhibitors such as LU99A and

SW1573 remained relatively resistant to RM-018. RMC-4998 had a similar pattern of inhibition to RM-018 in this panel of cell lines but with IC50 values approximately 10-fold lower reflecting its increased potency (Fig. 1b). To directly compare the effects of inactive-state and active-state KRAS[G12C] inhibition, the impact of RM-018 on ERK signaling was compared with adagrasib in the LU65 and H23 cell lines. Consistent with the mechanisms of action, 100 nM adagrasib maximally suppressed ERK signaling after 2–4 h of treatment, whereas RM-018 caused disappearance of the pERK signal on immunoblot after just 1 h of treatment (Fig. 1c). RM-018 also inhibited ERK signaling more durably than adagrasib with less reactivation of the pERK signal at 24 and 48 h of incubation (Fig. 1c).

To quantify output and pathway reactivation more comprehensively, we assessed the effects of these inhibitors on the transcriptional output of the ERK pathway[20]. ERK phosphorylation does not vary linearly with output because the expression of the ERK phosphatases DUSP1, 4 and 6 is ERK dependent. We used PCR to determine the effects of KRAS[G12C] inhibitors on the expression of nine ERK-regulated mRNAs as a function of time (Fig. 1d). The mRNAs encode regulators of pathway activation and feedback and transcription factors regulated by ERK signaling. They were chosen based on the dynamic range in expression after pathway inhibition and their relatively short half-life which would allow assessment at 24 h.

Both types of compounds significantly inhibited the expression of each of the mRNAs at 4 h, but, in every case, RM-018 was significantly more potent than adagrasib (Fig. 1d). Moreover, whereas significant rebound in expression occurred by 24 h in 8/9 mRNAs in H23 cells treated with adagrasib, in those treated with RM-018, rebound occurred in only 3/9 mRNAs and to levels below those of adagrasib. In LU65, the initial effect of RM-018 was much greater than that of adagrasib and at 24 h, expression of 7 mRNAs was less than 10% of baseline, while this was the case for only 2 of the mRNAs in adagrasib treated cells. These results are consistent with those obtained by immunoblotting and clearly show that the active-state RAS[G12C] inhibitor RM-018 is more potent and less susceptible to late rebound than the inactive-state KRAS[G12C] inhibitors. However, despite more potent and durable inhibition of the ERK pathway with RM-018, heterogeneity of response is still apparent across the panel of KRAS[G12C] cell lines.

### Sensitivity of NSCLC cell lines to KRAS[G12C] inhibitors is highly correlated with their inhibition of PI3K-AKT-mTOR signaling

To investigate additional variables that affect sensitivity to RAS inhibition, the impact of the inactive-state KRAS[G12C] inhibitor AZD8037 on effector pathways of RAS, RAF/MEK/ERK and PI3K/AKT/mTOR signaling, was assessed in the panel of cell lines. To facilitate comparison, cell lines with less than 50% viable cells compared to control at 1 μM AZD8037 were categorized as sensitive, whereas those for which more than 75% were viable were categorized as resistant and those in between were categorized as intermediate (Fig. 1a). After 2 h of treatment with AZD8037, pERK was markedly inhibited in all 12 tested cell lines (Fig. 2a). In most, inhibition persisted for at least 24 h, with no or only slight rebound. In five cell lines (HOP62, H1792, HCC44, SW1573, LU99), significant rebound occurred at 24 h. Of these, two were resistant and three were intermediate, suggesting the persistence of ERK inhibition is related to sensitivity. However, in the other cell lines with intermediate sensitivity (H23, H1373, H2030, Calu1), ERK was still potently inhibited at 24 h, suggesting that other variables play a role in determining sensitivity.

We observed that inhibition of PI3K/mTOR varied significantly among cell lines after treatment with AZD8037 (Fig. 2a, Supplementary Fig. 2a). In the sensitive cell lines, we observe sustained inhibition of the mTORC1 substrates p-p70S6K T389 and p4EBP1 S65 at 24 h, while pAKT S473 is inhibited at the earlier time points but with evidence of rebound at 24 h. In contrast, the intermediate and resistant cell lines exhibit no suppression of p4EBP1 S65 and weak suppression of

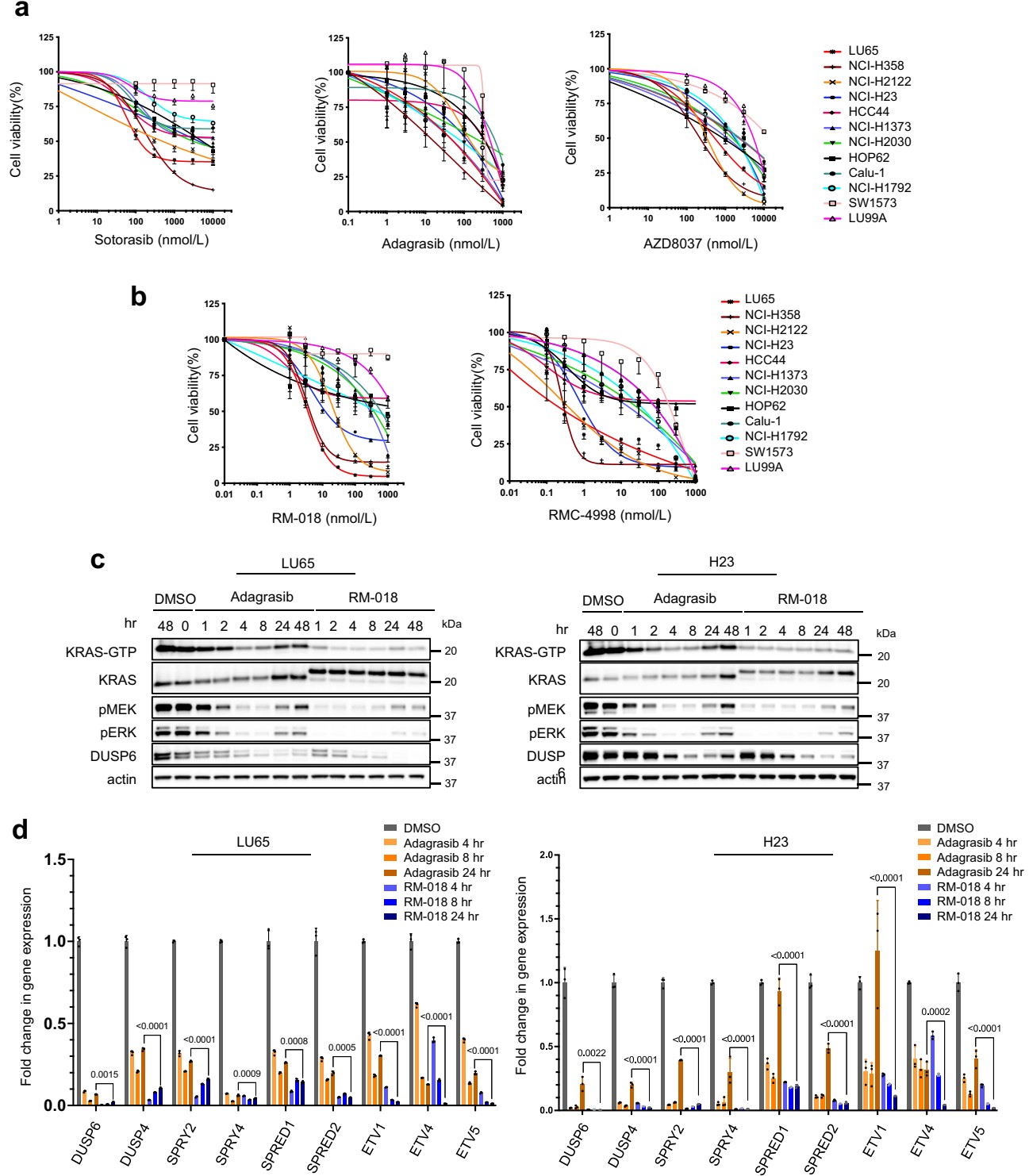

**Fig. 1 | Active-state KRAS^G12C inhibitors inhibit ERK pathway output more potently and durably than inactive-state KRAS^G12C inhibitors.** Cell viability data of KRAS^G12C mutant cell lines following treatment with (**a**) inactive-state KRAS^G12C inhibitors or (**b**) active-state KRAS^G12C inhibitors for 72 h. Data are the mean ± SD of *n* = 8 experimental replicates. **c** LU65 and H23 cell lines were treated with 100 nM Adagrasib or 100 nM RM-018 for the indicated times and lysates were analyzed by immunoblot. In parallel, the amount of active GTP-bound KRAS was determined by a RAS-GTP pulldown assay. **d** The mRNA expression of the indicated target genes was determined by quantitative PCR in LU65 and H23 cell lines following 100 nM Adagrasib or 100 nM RM-018 treatment for the indicated times. The data shown represent the means ± SD of *n* = 3 experimental replicates. *P* values are shown and were calculated by two-sided *t* test.

p-p70S6K T389, while pAKT S473 exhibits varying kinetics with initial inhibition followed by strong rebound in some cell lines. These data suggest that inhibition of PI3K/mTOR signaling, and suppression of 4EBP1 phosphorylation in particular, is strongly related to RAS inhibitor efficacy in these tumors. A similar pattern was observed with the active-state RAS^G12C inhibitor RM-018 (Supplementary Fig. 2b).

To provide quantification of this observation, we performed correlative analysis between inhibition of cell growth and inhibition of

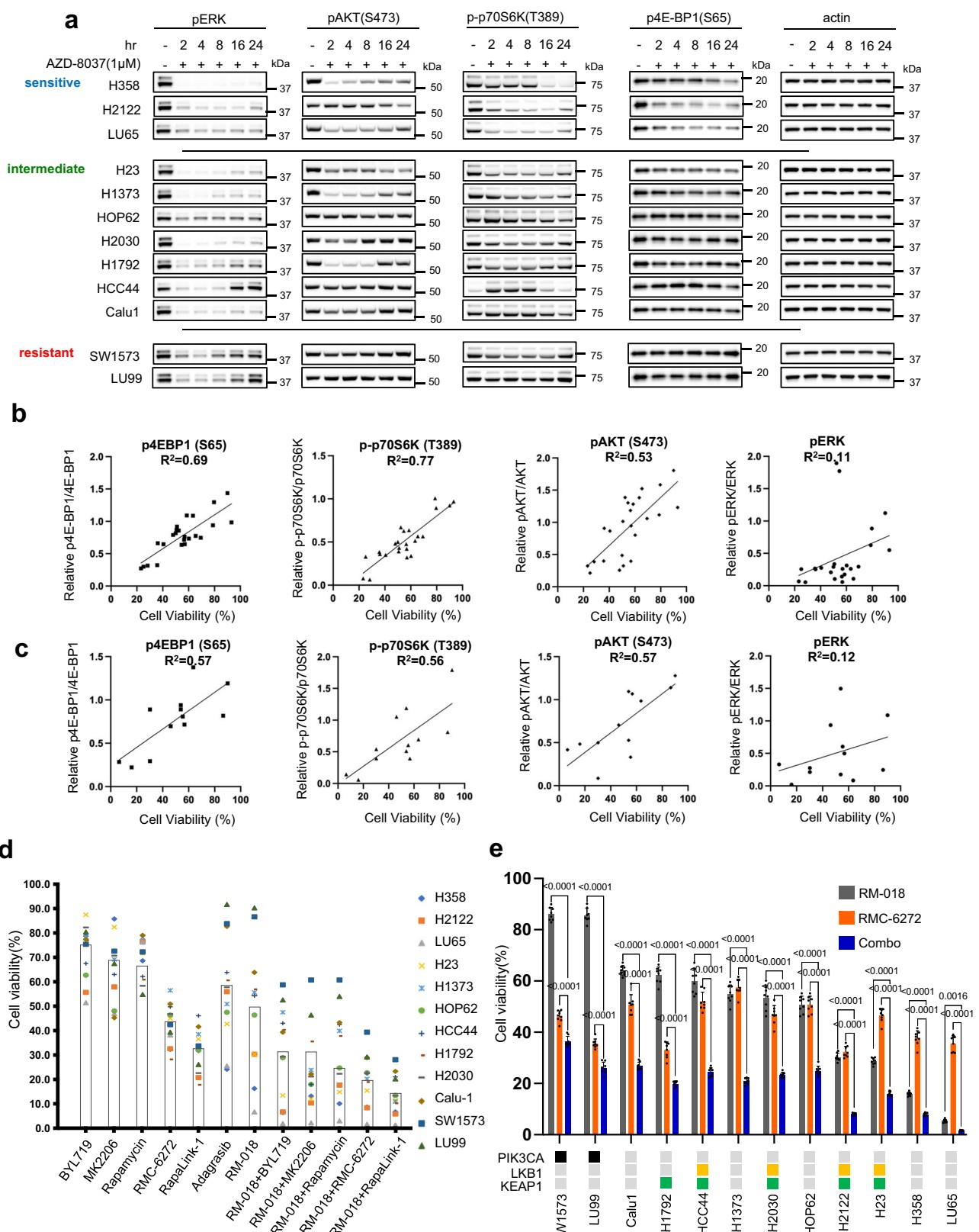

ERK and PI3K/mTOR signaling. We found that inhibition of cell growth by the inactive-state KRAS$^{G12C}$ inhibitors sotorasib and AZD8037 is highly correlated with the degree of inhibition of pAKT S473, p-p70S6K T389, and p4E-BP1 S65 with $R^2$ values ranging from 0.53–0.77, whereas pERK inhibition was not significantly correlated with an $R^2$ value of 0.11 (Fig. 1a, Fig. 2b, Supplementary Fig. 2a). This

analysis was also performed with RM-018, and a similar pattern of correlation was observed (Fig. 1b, Fig. 2c, Supplementary Fig. 2b). These results suggest that suppression of ERK signaling by KRAS$^{G12C}$ inhibitors is necessary but not sufficient to sensitize the cells; additional suppression of PI3K/mTOR signaling is also needed. This is consistent with previous reports and with our previous finding that

**Fig. 2 | Sensitivity of NSCLC cell lines to KRAS^G12C inhibitors is highly correlated with their inhibition of PI3K-AKT-mTOR signaling. a** Immunoblots of KRAS^G12C mutant cell lines treated with 1 μM AZD8037 for the indicated times. Scattered plot of cell viability for the KRAS^G12C mutant cell lines following 72 h treatment plotted against quantified immunoblot intensities of the indicated phosphorylated protein normalized against the respective total protein after 24 h treatment. The plots show 12 cell lines plotted twice each for independent experiments with either 1 μM Sotorasib or 1 μM AZD8037. $R^2$ calculated by linear regression. **c** Same as (**b**) but with 12 cell lines plotted once each after treatment with 100 nM RM-018. **d** Cell viability as percentage change from control of KRAS^G12C mutant cell lines treated for 72 h with BYL719 1 μM, MK2206 1 μM, Rapamycin 10 nM, RMC-6272 1 nM, RapaLink-1 10 nM, Adagrasib 100 nM, RM-018 100 nM, or combination treatments as indicated. Each point represents the mean of $n = 8$ experimental replicates, and the boxes represent the mean of the cell lines. **e** Cell viability as percentage change from control for KRAS^G12C mutant cell lines treated with either 100 nM RM-018, 1 nM RMC-6272 or the combination for 72 h. Data are means ± SD for $n = 8$ experimental replicates. $P$ values are shown and were calculated by one-way ANOVA with post-hoc Tukey's test. The status of genetic co-alterations is shown.

anti-proliferative efficacy requires dephosphorylation of 4E-BP1 and its consequent inhibition of cap-dependent translation[15]. At the time, this was accomplished by combining a MEK inhibitor with an AKT inhibitor, each of which inhibits signaling in both normal and tumor cells, however this combination was found to be intolerable in vivo. Our current data and the development of mutant allele-selective RAS inhibitors led us to revisit this idea.

The variability of inhibition of PI3K/mTOR signaling and its association with sensitivity suggests that the relationship may be causal and that combining the RAS inhibitor with a more potent inhibitor of PI3K or mTOR activity might significantly enhance antitumor activity. We therefore tested the effects of inhibitors of different components of the PI3K pathway, alone and in combination with RAS inhibition (Fig. 2d). We chose RM-018 for further analysis as a representative and potent active-state RAS^G12C inhibitor. For PI3K/mTOR pathway inhibitors, we utilized BYL719, a PI3K alpha-specific inhibitor, MK2206, an allosteric pan-AKT inhibitor, rapamycin, an allosteric mTOR inhibitor that preferentially inhibits mTORC1 but is a weak inhibitor of 4E-BP1 phosphorylation, and two third-generation potent bi-steric inhibitors of mTOR: RapaLink-1 and RMC-6272 are both bi-steric inhibitors comprised of a rapamycin-like moiety covalently linked to an ATP-competitive inhibitor of mTOR kinase[21–23]. RMC-6272 is relatively selective for mTORC1 while sparing mTORC2 over a significant concentration range, as compared to Rapalink-1 (Supplementary Fig. 2c, d), while both effectively inhibit 4E-BP1 phosphorylation in KRAS^G12C NSCLC cells[21]. RMC-6272 is a preclinical tool compound, representative of the investigational agent RMC-5552. The selective bi-steric mTORC1 inhibitors retain antitumor activity but do not cause hyperglycemia in preclinical models[23].

We observed that PI3K inhibition, AKT inhibition and rapamycin had modest effects on proliferation of cells in the panel, with an average 20–40% reduction in cell growth compared to control at 72 h (Fig. 2d). However, the bi-steric mTOR inhibitors Rapalink-1 and RMC-6272 were notably more potent with greater than 50% average reduction. We then tested each of these inhibitors in combination with RM-018. PI3K or AKT inhibition had modest additive benefits in combination with RM-018. However, combining RM-018 with the mTOR inhibitors had a more marked synergistic effect, with the bi-steric mTOR inhibitors demonstrating a greater than 80% average growth reduction across the cell line models. The combination of RM-018 and RMC-6272 showed a significant reduction in cell viability compared to either compound alone in every model tested, regardless of co-mutation status (Fig. 2e). These results suggest that in this panel of KRAS^G12C mutant NSCLC cell lines, mTORC1 inhibition in particular is critical in the context of combination treatment with the KRAS^G12C inhibitor. We were encouraged to further explore this therapeutic strategy in vivo in relevant preclinical models.

### The combination of KRAS^G12C and mTORC1 selective inhibition causes durable tumor regressions in KRAS^G12C mutant NSCLC models in vivo

We first tested the orally bioavailable active-state KRAS^G12C inhibitor RMC-4998 and the mTORC1-selective bi-steric inhibitor RMC-6272 in the cell-line derived H2122 KRAS^G12C lung adenocarcinoma xenograft model (CDX). RMC-6272 was administered once weekly by

intraperitoneal injection, and RMC-4998 was administered once daily by oral gavage. When administered as single agents, the mTORC1 inhibitor caused modest inhibition of tumor growth, while the KRAS^G12C inhibitor prevented growth for ~30–35 days after which tumors began to grow on treatment. In contrast, the combination of RMC-4998 with RMC-6272 caused almost complete tumor regression beginning soon after administration and persisting throughout the 50 days of treatment (Fig. 3a). The combination was well tolerated without weight loss and neither inhibitor monotherapy nor the combination caused significant hyperglycemia over 4 weeks of treatment (Supplementary Fig. 3a, b). We also tested the investigational candidates RMC-6291 (active-state RAS^G12C inhibitor)[19], and RMC-5552 (bi-steric mTORC1-seleive inhibitor)[23] in both the H2122 and H2030 CDX models (Fig. 3b, Supplementary Fig. 3c). In H2122, the responses to RMC-6291 and RMC-5552, alone and in combination, were very similar to those seen with their preclinical tool compounds, RMC-4998 and RMC-6272, respectively. In H2030, both single agents caused a modest slowing of tumor growth but in combination tumor regressions increased over time until there were minimal residual tumors on day 70.

We then extended our study of this combination therapy approach to a series of relatively insensitive KRAS^G12C mutant lung cancer PDX models with various co-existent genetic lesions (mutant *TP53, STK11/LKB1, KEAP1, MEK2*) (Fig. 3c). The combination of RMC-4998 with RMC-6272 achieved dramatic tumor shrinkage in three of the four models with clear synergy in LX349, and LX369. These responses were seen in all tumor-bearing mice harboring the LX349 and LX369 PDX models (Fig. 3d) and were durable over a period of seven weeks. In LX349, both the KRAS^G12C inhibitor and the combination were effective, but resistance to the KRAS^G12C inhibitor, and not the combination treatment group, developed on day 40. By contrast, both the KRAS^G12C inhibitor and the mTORC1 inhibitor caused only a slight growth delay in LX369, but the combination caused complete tumor regression by day 40, with continued response at day 75. Interestingly, LX699, the only model that did not undergo regression with the combination and grew slowly during the 36 days of treatment, harbored a *MEK2* G214W mutation, which while previously uncharacterized, is possibly a mechanism of KRAS inhibitor resistance in this model. None of the models developed resistance to the combination therapy while on treatment. No significant body weight loss was observed in any of the mice during the treatment period (Supplementary Fig. 3d).

### Combined inhibition of KRAS^G12C and mTORC1 blocks Cyclin D1 expression at the transcriptional and translational levels

We wished to further explore the mechanistic basis for the synergistic combination activity we observed. We have previously shown that the PI3K/AKT/mTOR and RAS/RAF/ERK pathways converge on key cellular processes, including Cyclin D-Cdk4/6 activation and cap-dependent translation and that both pathways must be inhibited to affect tumor growth or survival[15,24]. In H1373 and H2122, both RM-018 and RMC-6272 after 24 h treatment each resulted in a ~50% increase in cells in G0/G1 with a concomitant decrease in S phase cells. However, combination treatment led to a much more profound effect with a 110% increase in G0/G1 cells and concomitant decrease in S phase cells (Fig. 4a,

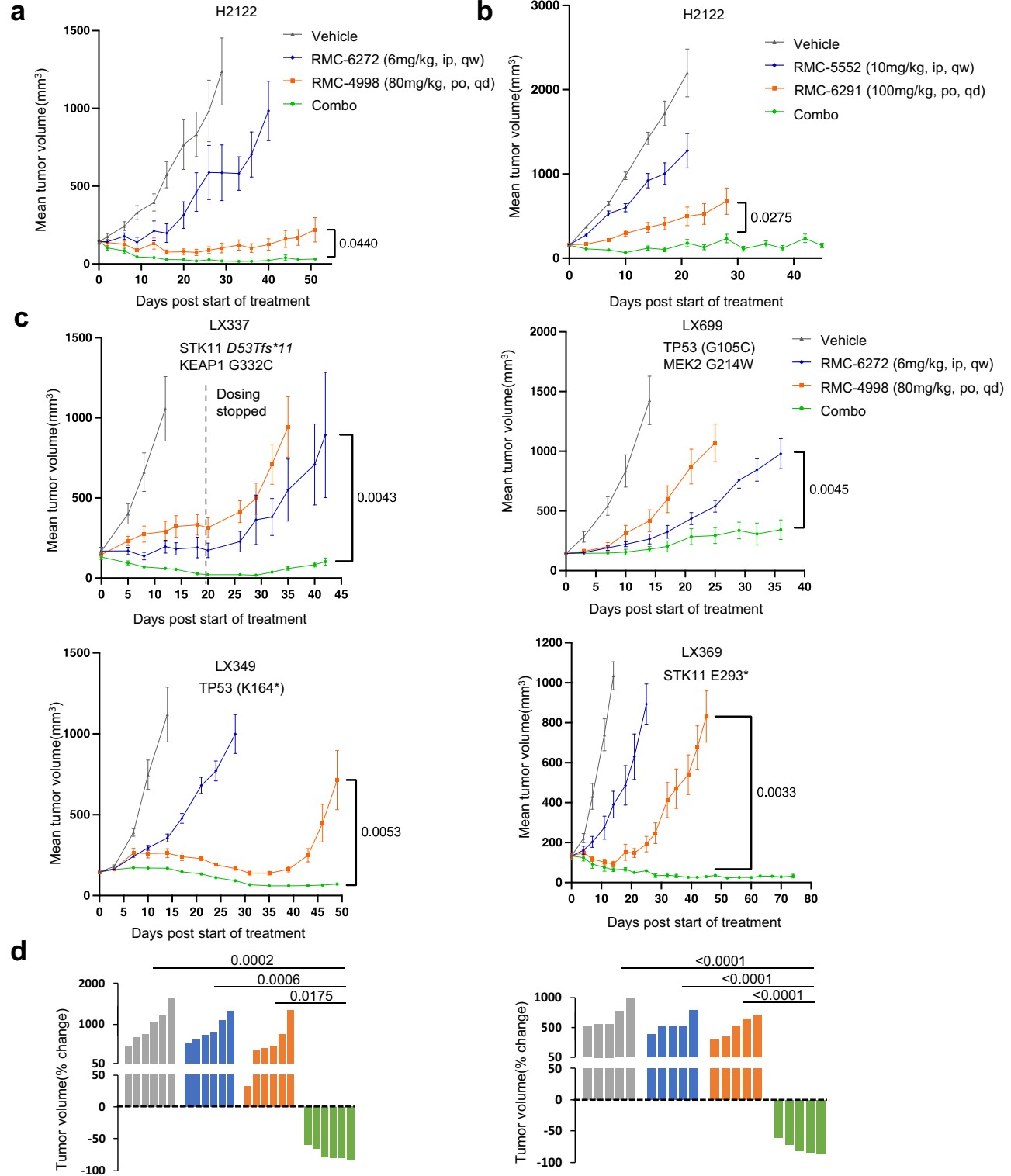

**Fig. 3 | The combination of KRAS$^{G12C}$ and mTORC1 selective inhibition causes durable tumor regressions in KRAS$^{G12C}$ mutant NSCLC models in vivo. a** H2122 xenografts were treated with vehicle, RMC-6272 6 mg/kg weekly, RMC-4998 80 mg/kg daily, or the combination. Tumor volumes were plotted from the start of treatment (mean ± SEM, n = 5 mice in each cohort). *P* values are shown and were calculated by two-sided t-test. **b** H2122 xenografts were treated 10 days post-implant with the vehicle, RMC-6291 100 mg/kg daily, RMC-5552 10 mg/kg weekly, or the combination. Tumor volumes were plotted over time from the start of treatment (mean ± SEM, n = 8 mice in each cohort). *P* values are shown and were

calculated by two-sided t-test. **c** Four different KRAS$^{G12C}$ mutant lung PDX tumors were implanted into NSG mice. PDXs were treated with the vehicle, RMC-6272 6 mg/kg weekly, RMC-4998 80 mg/kg daily, or the combination. Tumor volumes were plotted over time from the start of treatment (mean ± SEM, n = 5 mice in each cohort for LX349 and LX699, n = 6 mice in each cohort for LX369 and LX337). *P* values are shown and were calculated by two-sided *t* test. **d** Waterfall plot showing the percentage change in tumor volume (relative to initial volume) for individual LX349 (left) and LX369 (right) tumors at the end of treatment. *P* values are shown and were calculated by one-way ANOVA with post-hoc Tukey's test.

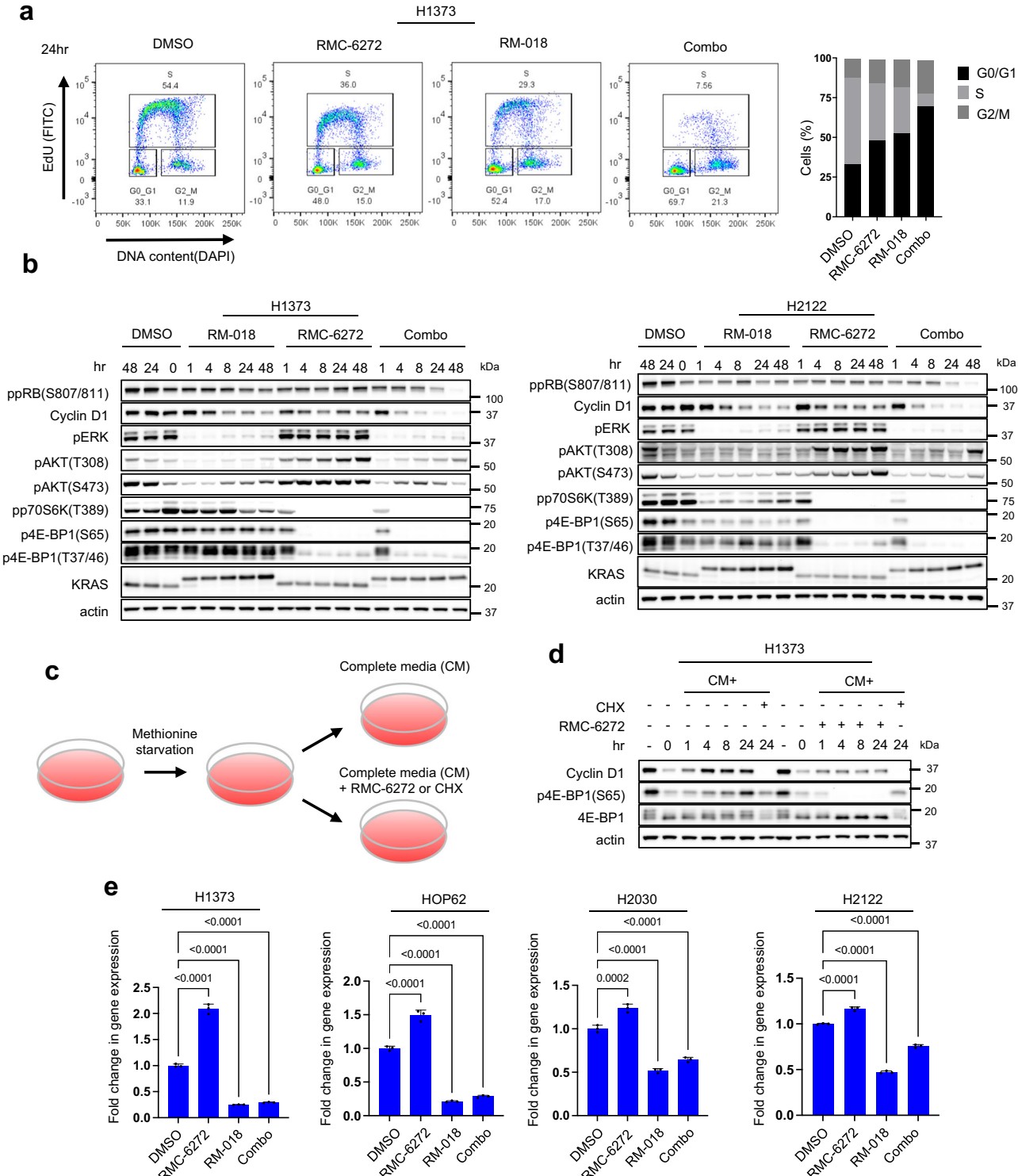

**Fig. 4 | Combined inhibition of KRAS^G12C and mTORC1 blocks Cyclin D1 expression at the transcriptional and translational levels. a** H1373 cells were treated with 1 nM RMC-6272, 100 nM RM-018, or the combination for 24 h. Cell cycle states were detected with EdU-DAPI-based flow cytometry. **b** H1373 and H2122 cells were treated with either 100 nM RM-018, 1 nM RMC-6272, or the combination for the indicated times, and lysates were analyzed by immunoblot. **c** Schema of the flow for the methionine starvation experiment in (**d**). **d** H1373 cells were cultured in methionine starved medium for 24 h, followed by adding back complete media (CM) alone, or with either 1 nM RMC-6272 or 50 μg/ml cycloheximide (CHX) for the indicated times, and lysates were analyzed by immunoblot. **e** Cyclin D1 mRNA levels were determined by quantitative PCR in H1373, HOP62, H2030, and H2122 cells following 1 nM RMC-6272, 100 nM RM-018, or the combination for 24 h. The data shown represent the means ± SD of n = 3 experimental replicates. *P* values are shown and were calculated by one-way ANOVA with post-hoc Tukey's test.

Supplementary Fig. 4a). We investigated the underlying mechanism of this phenomenon in the H1373, H2122, HOP62, and H2030 cell lines. In these models, each inhibitor alone caused a modest decrease in Cyclin D1 protein levels without a significant decrease in retinoblastoma (RB) phosphorylation (Fig. 4b, Supplementary Fig. 4b). By contrast, the combination caused marked decreases in both Cyclin D1 and phosphorylated RB protein levels. To further evaluate the functional importance of Cyclin D1, we performed siRNA knockdown of Cyclin D1 in H1373 cells and observed that this is sufficient to decrease RB phosphorylation and inhibit cell proliferation to a similar degree as the combination of RM-018 and RMC-6272 (Supplementary Fig. 4c, d). In addition, Cyclin D1 knockdown is sufficient to cause cycle arrest with a 3-fold increase in cells in G0/G1 and a concomitant 80% decrease in S phase cells (Supplementary Fig. 4e).

As Cyclin D1 translation is mTORC1 dependent[25], we assessed whether it is affected by RMC-6272. We starved cells for methionine for 24 h to inhibit initiation of translation, then refed cells 24 h later with complete media with or without RMC-6272 (Fig. 4c). The global translation inhibitor cycloheximide was used as a positive control. Methionine deprivation reduced Cyclin D1 protein levels while mRNA levels were not significantly inhibited (Fig. 4d, Supplementary Fig. 4f). Cyclin D1 protein levels immediately recovered once complete media was added back while RMC-6272 suppressed this increase (Fig. 4d). These data suggest that RMC-6272 inhibits induction of Cyclin D1 protein expression primarily via blocking its translation. However, we noted that RMC-6272 alone does not completely inhibit Cyclin D1 expression, and thus assessed whether RM-018, RMC-6272 or the combination affected the levels of Cyclin D1 mRNA. Across four cell line models, Cyclin D1 mRNA levels increased by ~50–100% following RMC-6272 treatment, while it was suppressed with RM-018 treatment by ~50–75% (Fig. 4e). Combination treatment also suppressed Cyclin D1 mRNA to approximately the same level as RM-018 alone, demonstrating that the effect of RM-018 dominates in this setting. In sum, these results suggest that the effects of RMC-6272 on Cyclin D1 translation are buffered by ERK-dependent induction of Cyclin D1 mRNA. The combination of RM-018 and RMC-6272 blocks both processes and synergistically reduces cellular Cyclin D1 protein levels.

## Combined inhibition of KRAS[G12C] and mTORC1 kinase synergize to induce apoptosis by inducing BIM and reducing MCL-1 expression

The in vivo regression of KRAS[G12C] NSCLC murine models suggests that the combination therapy induces cell death. We asked whether mTORC1 inhibition enhances KRAS[G12C] inhibitor-induced cytotoxicity. Induction of death in the KRAS[G12C] mutant H1373, H2122, and HOP62 cell lines was assessed with the annexin V-propidium iodide (PI) assay following 72 h of treatment with RM-018, RMC-6272, or the combination. In H1373, H2122, and HOP62, RM-018 alone approximately doubled the percentage of annexin V positive cells, while RMC-6272 only caused a minimal increase (Fig. 5a, Supplementary Fig. 5a). However, compared to RM-018 treatment alone, the combination with RMC-6272 significantly enhanced total cell death by three to four-fold as compared to controls. Induction of death by the combination is suppressed by the pan-caspase inhibitor Z-VAD-FMK (Z-VAD) in H1373 and H2122 cells to varying degrees, suggesting that a significant proportion of the death is due to activation of the caspase cleavage cascade, but that other mechanisms may be involved as well (Supplementary Fig. 5b). Consistent with the results of annexin V- PI assay, the inhibitor combination induced a significant time-dependent increase in cleaved PARP compared to RM-018 alone (Fig. 5b). Notably, Cyclin D1 knockdown alone did not induce cleaved PARP or increase the percentage of annexin V positive cells (Supplementary Fig. 5c).

Binding of the BH3 domain-only proteins such as BIM, PUMA, and BID to BCL2 family members like MCL-1 inhibits the anti-apoptotic effects of the latter and induces apoptosis[26]. MCL-1 translation has

previously been shown to be mTOR dependent[27,28]. Thus, we further investigated if the effect of mTORC1 inhibition on MCL-1 translation was the basis for the enhanced cell death in combination with the KRAS[G12C] inhibitor. We found that RMC-6272 decreased MCL-1 protein levels, in agreement with the previous studies, but alone does not induce PARP cleavage (Fig. 5b). MCL-1 protein levels were reduced following methionine deprivation and recovered following addition of complete media, whereas RMC-6272 blocked this recovery (Fig. 5c). In contrast, mRNA levels of MCL-1 were induced by methionine starvation, demonstrating that the reduction in protein level is due to the inhibition of MCL-1 translation (Supplementary Fig. 5d). On the other hand, RM-018 had little effect on MCL-1 protein levels but strongly induced the pro-apoptotic BH3-only proteins BIM and PUMA (Fig. 5b). Inhibition of ERK-dependent phosphorylation of BIM stabilizes the latter and accounts for increased expression of the protein[29]. In H2122, addition of RM-018 further reduced MCL-1 protein levels compared to RMC-6272 treatment alone. As a possible mechanism for this observation, it has been previously reported that phosphorylated BIM binds to and stabilizes MCL-1[30].

From these data, we hypothesized that induction of apoptosis by combined inhibition of KRAS[G12C] and mTORC1 is in part due to inhibition of MCL-1 translation by the latter and induction of BIM and/or PUMA protein expression by the former. We therefore knocked down MCL-1 with siRNA (Supplementary Fig. 5e) and found that it was sufficient to cooperate with RM-018 inhibition to enhance caspase3/7 activity in the H1373, H2122, and HOP62 cell lines (Supplementary Fig. 5f). Moreover, siRNA knockdown of BIM, but not PUMA, significantly reduced induction of PARP cleavage by the combined inhibition of KRAS[G12C] and mTORC1 in H1373 cells (Fig. 5d). This suggests that in these cells, BIM is necessary for apoptosis induction, while PUMA is dispensable. MCL-1 overexpression also prevented the induction of PARP cleavage by the combination, suggesting that MCL-1 is sufficient in these cells to inhibit apoptosis (Fig. 5e). We demonstrated a similar effect in H2122 xenograft tumors in vivo 48 h after dosing with RMC-4998 and RMC-6272. The combination effectively suppresses Cyclin D1 and MCL-1 protein levels, and 4E-BP1 phosphorylation (Fig. 5f). As predicted, immunohistochemistry analysis shows that the combination of the clinical candidates RMC-6291 (active-state KRAS[G12C] inhibitor) and RMC-5552 (bi-steric mTORC1-selective inhibitor) similarly synergistically induces Caspase 3 cleavage (Fig. 5g, h, Supplementary Fig. 5g), extending our findings to the investigational agents under clinical evaluation.

Taken together, these data demonstrate that BIM induction by KRAS[G12C] inhibition and block of MCL-1 protein translation by mTORC1 inhibition together are important to induce the cell death needed for the antitumor activity of the combination treatment.

## KRAS[G12C] inhibition and mTORC1 inhibition cause synergistic reduction in cap-dependent translation

We have previously shown that AKT and MEK inhibition cooperate to inhibit 4E-BP1 phosphorylation and that dephosphorylation of 4E-BP1 is required for the anti-tumor activity of the combination[15]. eIF4E recognizes the mRNA m7GTP 5′ cap structure and assembles the eIF4F translation initiation complex which also includes the proteins eIF4A and eIF4G[31]. Complex assembly is inhibited by binding of 4E-BP1 to eIF4E, whereas phosphorylation of 4E-BP1 by mTORC1 releases eIF4E and allows cap-dependent translation to proceed[32]. Thus, RMC-6272 attenuates cap-dependent translation initiation via potent inhibition of the phosphorylation of 4E-BP1 by mTORC1 (Fig. 4b, Supplementary Fig. 2c-d).

Beads conjugated to the 5′cap structure m7GTP were able to pull down eIF4G, eIF4A, and eIF4E from lysates of the H1373 and H2122 cell lines. As expected, the eIF4E/eIF4G interaction was reduced by knockdown of eIF4E, thus suppressing formation of the translation initiation complex (Supplementary Fig. 6a). In the H1373 and H2122

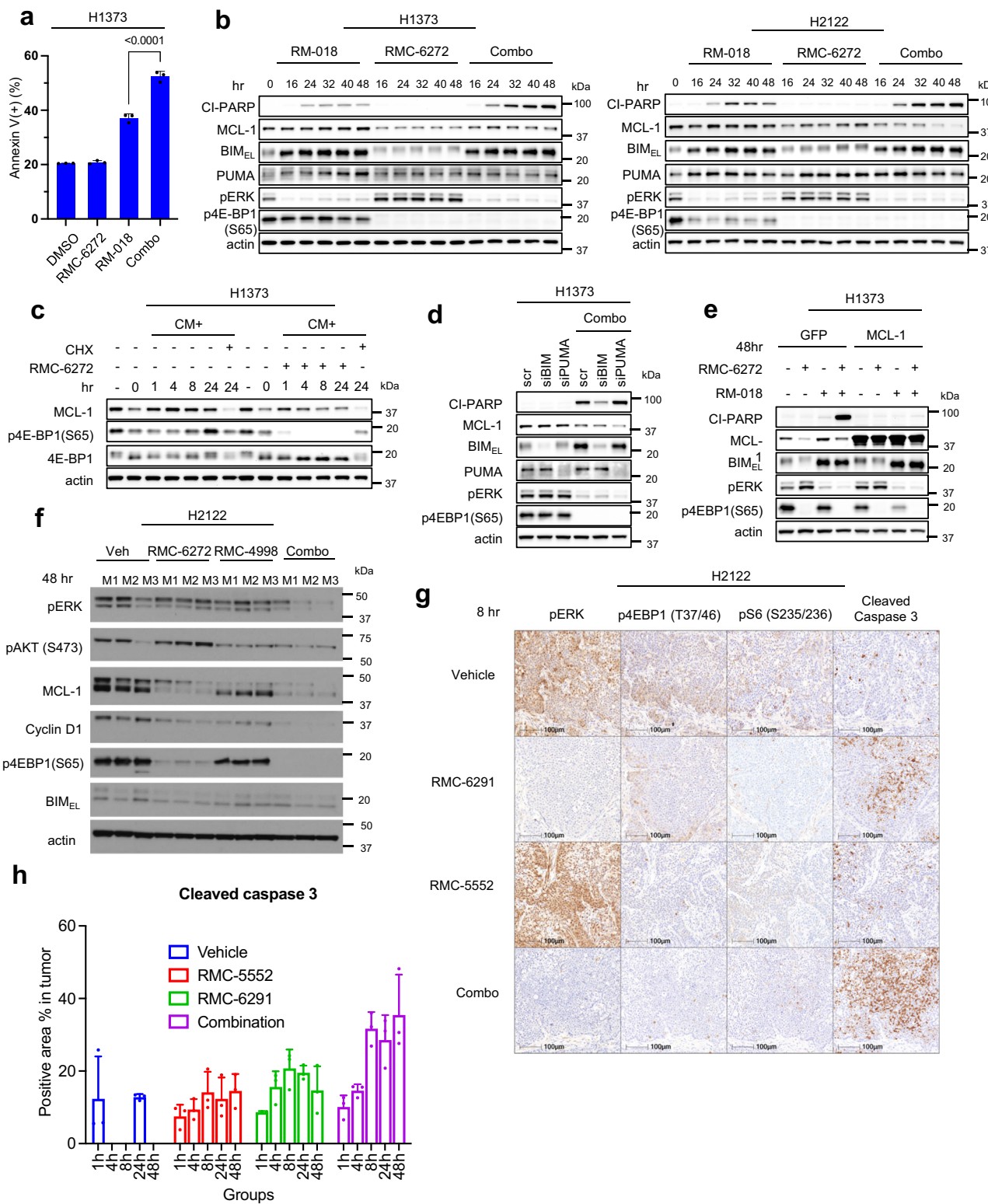

models, eIF4E knockdown alone had a minimal effect on apoptosis, whereas treating these cells with RM-018 caused an approximate two-fold increase in Annexin V+ cells (Fig. 6a). These results suggest that the effect of RMC-6272, through inhibition of 4E-BP1 phosphorylation, is similar to that of knocking down eIF4E, with both leading to inhibition of eIF4F translation initiation complex assembly. Alone neither causes apoptosis, however, either knockdown of eIF4E or RMC-6272 treatment cooperates with KRAS$^{G12C}$ inhibition to induce significant cell death. Consistent with this mechanism, knockdown of eIF4E inhibited

MCL-1 and Cyclin D1 expression in H1373 cells, and to a lesser degree, in H2122 cells (Fig. 6b). Combining RM-018 with eIF4E knockdown caused greater inhibition of expression of both Cyclin D1 and MCL-1 than either alone. S6K phosphorylation was induced by knockdown of eIF4E whereas it was inhibited by RM-018. S6K affects the helicase activity of eIF4A through other translation regulators such as eIF4B and PDCD4[33]. Therefore, it is possible that the dual inhibition of 4E-BP1 and S6K phosphorylation, as observed with RMC-6272, may contribute to the inhibition of cap-dependent translation.

**Fig. 5 | Combined inhibition of KRAS^G12C and mTORC1 kinase synergize to induce apoptosis by inducing BIM and reducing MCL-1 expression. a** H1373 cells were treated with 100 nM RM-018, 1 nM RMC-6272, or the combination for 72 h, and analyzed by FACS to quantify annexin V positive cells. Data are means ± SD for n = 3 experimental replicates. *P* values are shown and were calculated by one-way ANOVA with post-hoc Tukey's test. **b** Immunoblot analysis of lysates from H1373 and H2122 cells treated with 100 nM RM-018, 1 nM RMC-6272, or the combination for the indicated times. **c** H1373 cells were cultured in methionine-starved medium for 24 h, then replaced with complete media (CM) alone, or with either 1 nM RMC-6272 or 50 µg/ml cycloheximide (CHX) for the indicated times, and lysates were analyzed by immunoblot. **d** H1373 cells were transfected with siRNAs targeting BIM, PUMA or scramble (scr) siRNA and incubated for 24 h. The cells were then treated with the combination of 1 nM RMC-6272 and 100 nM RM-018 for 48 h and lysates

were analyzed by immunoblot. **e** H1373 cells infected with a GFP control or MCL-1 expressing lentiviral plasmid were treated with 100 nM RM-018, 1 nM RMC-6272, or the combination for 48 h and lysates were analyzed by immunoblot. **f** H2122-derived xenografts were treated with vehicle, RMC-6272 6 mg/kg once, RMC-4998 80 mg/kg x 2 doses 24 h apart, or the combination. 48 h after start of treatment, the tumors were collected, lysed and immunoblotted with the indicated antibodies. M1, M2, M3 represent individual mice from each treatment condition. **g** IHC analysis of H2122-derived xenograft tumors collected 8 h after treatment with vehicle, RMC-5552 (10 mg/kg, ip), RMC-6291 (100 mg/kg, po), or the combination. The magnification of all IHC slides are 40x, scale bars are 100 µm. **h** Quantification of Cleaved Caspase 3 IHC analysis shown in (g). The data shown represent the means ± SD of n = 3 experimental replicates.

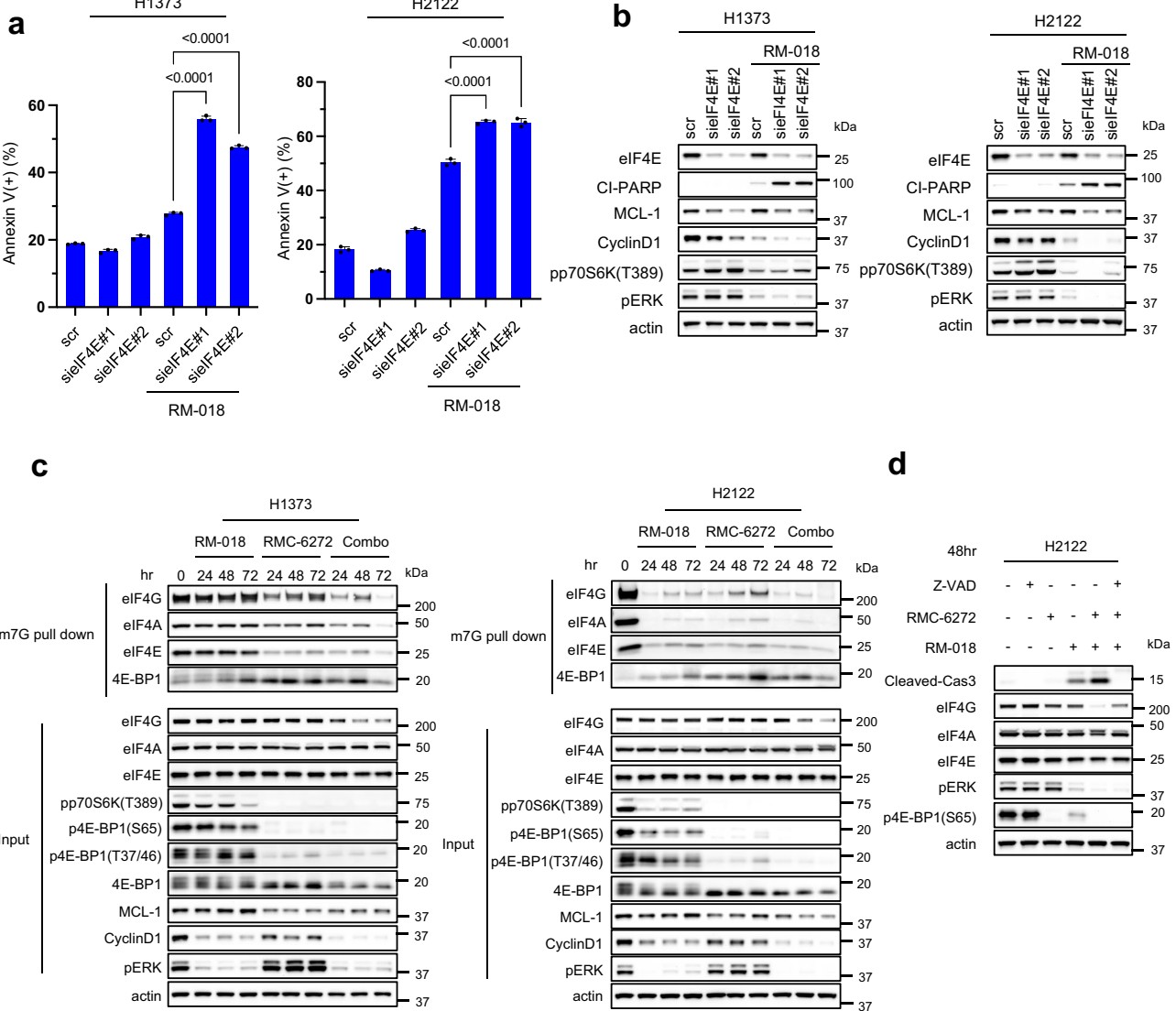

**Fig. 6 | KRAS^G12C inhibition and mTORC1 inhibition cause synergistic reduction in cap-dependent translation. a** H1373 and H2122 cells were transfected with two different siRNAs targeting eIF4E or scramble (scr) siRNA and cultured for 24 h. Media was replaced with or without 100 nM RM-018 and cells were treated for an additional 48 h and analyzed by FACS to quantify annexin positive cells. Data are means ± SD for n = 3 experimental replicates. *P* values are shown and were calculated by one-way ANOVA and post-hoc Tukey's test. **b** H1373 and H2122 cells were transfected with two different siRNAs targeting eIF4E or scramble (scr) siRNA and

cultured for 24 h. Media was replaced with or without 100 nM RM-018 and cells were treated for an additional 48 h, and lysates were analyzed by immunoblot. **c** H1373 and H2122 cells were treated with 1 nM RMC-6272, 100 nM RM-018, or the combination for the indicated times, followed by incubation with m⁷GTP-conjugated beads, and lysates were probed by immunoblot in parallel with whole cell extracts (input). **d** H2122 cells were treated with 100 µM Z-VAD-FMK, 1 nM RMC-6272, 100 nM RM-018, or indicated combinations for 48 h, and lysates were analyzed by immunoblot.

We assessed the effects of the RM-018 and RMC-6272 combination treatment on the formation of the cap-dependent translation initiation complex in the H1373 and H2122 cell lines. As expected, the association of 4E-BP1 with eIF4E was enhanced upon RMC-6272 treatment, resulting in inhibition of eIF4F complex assembly (Fig. 6c). eIF4F assembly was further inhibited with the combination. We observed the expression of total eIF4G was reduced by the combination in H1373 whereas KRAS[G12C] inhibition is sufficient in H2122. eIF4G has caspase sites and can be cleaved by caspase-3 during apoptosis[34]. Z-VAD prevented loss of eIF4G expression by the combination suggesting that activation of caspases including caspase-3 by the combination treatment also induces cleavage of eIF4G (Fig. 6d). Thus, we conclude that induction of apoptosis by inhibiting translation and inducing BIM causes a feedforward loop causing further inhibition of translation through eIF4G degradation. This may likely contribute to the antitumor effects of KRAS[G12C] inhibitors alone or in combination with mTORC1 inhibition.

## Discussion

The overall response rate of inactive-state KRAS[G12C] inhibitors sotorasib and adagrasib in patients with *KRAS[G12C]* mutant NSCLC is 37-43%, a major advance albeit with significant room for improvement[7,8]. We investigated a class of active-state RAS[G12C] inhibitors which forms a tri-complex with the abundant intracellular protein cyclophilin A and with GTP-bound KRAS[G12C] to which a covalent bond is formed[19]. We demonstrate that representative compounds of this class, RM-018 and RMC-4998, are more potent than current inactive-state KRAS[G12C] inhibitors in a significant subset of the NSCLC cell line models studied, consistent with the differentiated mechanism of action. We further show that RM-018 leads to more potent inhibition of ERK output and less feedback reactivation than the inactive-state KRAS[G12C] inhibitors.

However, even with more potent inhibition of the pathway, we still observe significant heterogeneity in response to active-state RAS[G12C] inhibitors in a panel of NSCLC cell line models. While previous studies have characterized acquired resistance to KRAS[G12C] inhibition and identified causal mutations and non-genetic mechanisms including RTK activation and epithelial-to-mesenchymal transition[11-14,35], the mechanisms underlying differential sensitivity to KRAS[G12C] inhibition are less well understood. In the present study, our data demonstrate that there is a strong correlation between sensitivity to KRAS[G12C] inhibition and the degree of PI3K/mTOR pathway inhibition, particularly inhibition of mTORC1 targets including 4E-BP1 and S6K. This correlation persists for sensitivity to active-state KRAS[G12C] inhibitors, suggesting that even with more potent inhibition of KRAS[G12C], some cells have inherent KRAS[G12C]-independent PI3K pathway activation. We identify three types of cell lines: 1) those in which both ERK and PI3K signaling are potently inhibited by the KRAS[G12C] inhibitor, 2) those in which neither is, and 3) those in which ERK signaling is sensitive and PI3K signalling is less and variably sensitive. The mechanistic basis for this phenomenon is not clear, but we speculate that in the third group of cells, PI3K/mTOR signaling could be entirely KRAS[G12C]-independent, or driven by both mutant and mutant-independent pathways. The latter could include activation by wild-type RAS mechanisms. An additional layer of complexity is the cross-regulation between the two pathways with ERK signaling feeding into several nodes of PI3K/mTOR such as its phosphorylation of TSC2[36-38]. Nevertheless, these data are consistent with our previous findings that inhibition of the growth of tumors in which ERK and PI3K/mTOR signaling are dysregulated requires inhibition of both pathways[15,24].

We, therefore, tested whether inhibitors of individual components of the PI3K pathway affected the growth inhibition or induction of cell death by KRAS[G12C] inhibitors. PI3K, AKT, and mTOR kinase inhibitors all synergized with KRAS[G12C] inhibition to varying degrees, with mTORC1 kinase inhibitors being most efficacious. These data support the mechanistic importance of our findings that antitumor activity of KRAS[G12C] inhibitors is closely correlated with PI3K/mTOR pathway inhibition, especially inhibition of phosphorylation of mTORC1 substrates 4E-BP1 and S6K. These data are consistent with our previous findings that 4E-BP1 is a convergent target of PI3K/AKT/mTOR and RAF/MEK/ERK pathways and integrates their function at the level of regulation of cap-dependent translation[15]. Previous studies by other groups have also demonstrated the importance of mTORC1 inhibition for the effectiveness of MAPK or PI3K-pathway directed therapies in diverse tumor-types including lung, breast and melanoma[39-42]. Our data now further suggests the possibility that combined inhibition of KRAS[G12C] and mTORC1 might be useful in treating KRAS[G12C] mutant NSCLC. Previous strategies combining MEK and AKT inhibition proved to be poorly tolerated and not appropriate for treatment of patients[43,44]. In the present study, we utilize the bi-steric mTORC1-selective inhibitor RMC-6272 which demonstrates the strongest synergy with RAS inhibition, compared to PI3K, AKT, and allosteric mTORC1 inhibitors. We believe that substitution of the MEK inhibitor, which suppresses ERK signaling in normal and tumor cells and has a narrow therapeutic index, with a mutant-selective RAS inhibitor and the use of a mTORC1-selective kinase inhibitor, may allow the safe administration of this combination in humans.

We tested combined inhibition of KRAS[G12C] and mTORC1 in vivo and observed striking antitumor activity in a variety of CDX and PDX models of *KRAS[G12C]* mutant NSCLC with a variety of co-mutations. These responses are associated with deep regression in almost all cases and are markedly durable. We observed minimal body weight loss in animals with this combination treatment, and no hyperglycemia which has been a previous on-target adverse effect of PI3K/mTOR pathway inhibitors that target mTORC2[45]. These data suggest that the combination of a KRAS[G12C] inhibitor with an mTORC1-selective inhibitor is a promising therapeutic strategy for a wide subset of KRAS[G12C] mutant lung cancers warranting further study.

We have previously shown that several downstream processes are under the control of both ERK and PI3K/mTOR signaling and that their effective inhibition requires inhibition of both pathways[15,24]. Inhibition of either PI3K/mTOR or ERK signaling often relieves feedback inhibition of the other and causes its activation[46,47]. In this way the effects of inhibiting ERK or PI3K signaling are buffered by reactivation of the other pathway in both tumor and normal cells. We have used the KRAS[G12C] inhibitors to explore the effects of KRAS[G12C] activation in tumor models and identified three key cellular processes activated by mutant KRAS, effective inhibition of which requires inhibition of both PI3K/mTOR and ERK signaling: Cyclin D1 expression, cell survival, and cap-dependent translation. Inhibition of mTORC1 inhibits translation of Cyclin D1, but also induces expression of Cyclin D1 mRNA, blunting the effects of the former. Combined inhibition of both is required to maximize Cyclin D1 inhibition and cell cycle arrest. Conversely, the mechanism suggests that, by activating both PI3K and ERK signaling, mutant KRAS[G12C] amplifies cell cycle deregulation. Although mTORC1 inhibition reduces the translation of MCL-1, survivin and other elements of the antiapoptotic network, it does not by itself induce significant cell death. ERK inhibition by the KRAS[G12C] inhibitors induces pro-apoptotic-BH3 domain only proteins like BIM and PUMA but also, by itself, does not induce significant cell death. This provides a safety mechanism for both normal and tumor cells, whereby inhibiting either pathway alone does not cause the death of cells. However, tumor specific inhibition of the two pathways provides both signals and causes significant cell death. Similarly, cap-dependent translation is necessary for the transformed phenotype and we have shown previously that dephosphorylation of PTEN, which inhibits cap-dependent translation, is necessary for combined therapy to work[48]. Inhibition of both pathways is usually required for this to happen, but we did identify cell lines in which KRAS[G12C] inhibition alone is sufficient for maximal inhibition. These turn out to be the same cell lines in which KRAS[G12C] inhibition by itself most potently suppresses mTOR activity,

presumably via intact crosstalk mechanisms. Interestingly, while mutant selective KRAS[G12C] inhibition is sufficient to drive tumor regressions in such models in vivo, the combination was required to maintain durable responses.

Further questions remain such as the molecular determinants of the degree of RAS dependency of PI3K/mTOR signaling in different lineages and individual tumors. PI3K mutations are relatively common in colorectal cancer, but there are very few pancreatic or lung KRAS mutant tumors harboring genetic mechanisms for PI3K pathway activation. Much remains to be investigated regarding the complexity of regulation of cell survival networks or translational initiation in these tumors or the role of other coexistent genetic mutations. In summary, we have identified important features of mutant RAS inhibition in NSCLC that speak to its role in driving these tumors and the molecular consequences of the therapeutic inhibition thereof. We have determined the mechanistic rationale for combined inhibition of mTORC1 and mutant KRAS[G12C]. The preclinical data results described herein support the clinical evaluation of this combinatorial therapeutic strategy in patients with KRAS[G12C] mutant-driven NSCLC.

## Methods

### Cell culture and reagents

The lung cancer cell lines NCI-H358, NCI-H2122, NCI-H2030, HCI-H1373, NCI-H1792, NCI-H23, HOP62, Calu-1, and SW1573, A549 were purchased from the American Type Culture Collection. PC-9 was provided by Emily Cheng Lab (MSKCC). HCC44 was obtained from Korean cell line bank. LU-65 and LU-99A were obtained from the Japanese Cell Research Bank (Osaka, Japan). NCI-H358, NCI-H2122, NCI-H2030, HCI-H1373, NCI-H1792, NCI-H23, HOP62, SW1573, LU-65 and LU-99A cells were cultured in RPMI1640; A549 was cultured in DMEM; Calu-1 was cultured in McCoy's 5A. All media was supplemented with 2 mM glutamine, and 10% fetal bovine syndrome. All cells were maintained at 37 °C in 5% CO2. Cells were regularly screened for Mycoplasma using a MycoAlert Mycoplasma Detection Kit (Lonza). Sotorasib, Adagrasib, BYL719, Rapamycin, MK2206 were obtained from Selleck Chemical (Houston, TX). AZD8037 was obtained from AstraZeneca. RapaLink-1 was purchased from Med-ChemExpress. RM-018, RMC-6272, RMC-4998, RMC-6291, and RMC-5552 were provided by Revolution Medicines. Compounds were dissolved in DMSO to a final concentration of 10 mmol/l and stored at −20 °C.

### Cell proliferation analysis

For dose-response assay of cell viability, $2-3 \times 10^3$ cells were seeded in each well of 96-well plates and incubated overnight. Then cells were treated with increasing concentration of the indicated drugs for 72 h. Cell viability assay was performed using the Cell Titer-Glo luminescent cell viability assay (Promega, Madison, WI) in accordance with the manufacturer's instructions. Luminescence was measured on a SpectraMax M5 plate reader (Molecular Devices). The data were graphically displayed using GraphPad Prism (Ver.9.41).

### Immunoblotting

Cells were lysed with RIPA buffer (Cell Signaling Technologies, #9803), supplemented with protease and phosphatase inhibitors (Pierce Chemical, Thermo Fisher Scientific). Lysates were briefly sonicated and cleared by centrifugation at $18,407 \times g$ for 5 min at 4 °C. The supernatant was collected, and protein concentration was measured by the BCA kit (Pierce).

Xenograft tumors were homogenized in SDS lysis buffer (50 mM Tris-HCL pH 7.4, 10% Glycerol, 2% SDS) and boiled at 95 °C for 5 min. Lysates were then briefly sonicated, boiled again for 5 min, and cleared by centrifugation at $18,407 \times g$ for 10 min at room temperature. The supernatant was collected and protein concentration was determined using the BCA kit (Pierce). The protein samples were mixed with an appropriate volume of SDS-PAGE sample buffer and incubated at 70 °C for 10 min.

Equal amounts of protein (20 μg) were electrophoresed on a NuPAGE Bis-Tris protein gel (Invitrogen), and then transferred to nitrocellulose or PVDF membranes (Bio-Rad, Thermo Fisher Scientific) before blocking for 1 h at room temperature and incubating with primary antibody overnight at 4 °C. Membranes were then incubated with the appropriate secondary antibody, and detected by chemiluminescence with ECL detection reagents (Thermo Fisher Scientific, Millipore).

Primary antibodies obtained from Cell Signaling Technologies and used at 1:1000 dilution: pERK (#4370), ERK (#4696), pAKT T308 (#2965), pAKT S473 (#4060), AKT (#9272), p4EBP1 S65 (#9451), p4EBP1 T37/46 (#9459), 4EBP1 (#9452), p-p70S6K (#9234), p70S6K (#34475), pCRAF C338 (#9427), pMEK (#9154), CyclinD1 (#55506), ppRB (#8516), MCL-1 (#5453), BIM (#2933), PUMA (#12450), cleaved PARP (#5625), cleaved Caspase-3 (#9661), eIF4G (#2498), eIF4A (#2013), eIF4E (#2067), pEIF4E (#9741), Actin (#4970), BCL-XL (#2764). KRAS primary antibody was obtained from LSBio (#LS-C1765665). Signal was detected using iBright™ 1000 Imaging Systems (Thermo Fisher Scientific). All Western blot experiments were repeated at least twice, and a representative result is shown.

### m7GTP binding assay

Cells were homogenized in lysis buffer (50 mM HEPES-KOH (pH 7.4), 2 mM EDTA, 10 mM pyrophosphate, 10 mM b-glycerophosphate, 40 mM NaCl, 1% Triton X-100), supplemented with protease and phosphatase inhibitors (Pierce Chemical, Thermo Fisher Scientific). Cell extracts (250 μg protein) were incubated with 50 μl of m7GTP-sepharose beads (AC-155, Jana Bioscience, Germany) for 2 h in a rotary suspension mixer at 4 °C. The beads were washed three times with lysis buffer, then the bound protein was mixed with an appropriate volume of SDS-PAGE sample buffer and incubated at 70 °C for 10 min. After a brief centrifugation, supernatants were electrophoresed on a NUPAGE gel, and the proteins transferred onto a nitrocellulose membrane and immunoblotted.

### RAS activity assay

RAS activation assay was performed using Active Ras Pull-Down and Detection Kit (Thermo Fisher Scientific). To identify RAS-GTP, cell lysates were immunoprecipitated with a GST-fusion protein of the Ras-binding domain (RBD) of Raf1 along with glutathione agarose resin according to the manufacturer's protocol. The RAS activity assay was repeated at least twice and a representative result is shown.

### siRNA knockdown

Cells were seeded into 6-well plates at a density of $1-2 \times 10^5$ cells/well. 24 h later, cells were transfected with 20 nM ON-TARGET plus siRNA against BIM, PUMA, MCL-1, eIF4E, Cyclin D1 (Horizon Discovery), Select siRNA for Negative Control no.1 (Invitrogen) using Lipofectamine RNAi MAX (Invitrogen) according to the manufacturer's instructions. Knockdown was confirmed by western blotting analysis.

### Cell viability and caspase 3/7 activity assay

Activity of caspase 3/7 was analyzed by Caspase-Glo 3/7 Assay System (Promega) according to manufacturer's protocol. Briefly, $2 \times 10^3$ cells were seeded in each well of 96-well plates and incubated overnight. The cells were transfected with siRNAs MCL-1 or scramble siRNA and cultured for 24 h. Then, media was replaced with or without 100 nM RM-018 and cells were treated for an additional 24 h. At the end of experiment, the same volume of caspase-Glo 3/7 reagent was added in 96-well plates and incubated at room temperature in the dark for 1 h. Luminescence was measured on a SpectraMax M5 plate reader (Molecular Devices). Treatments were performed 8 times (n = 8). The data were graphically displayed using GraphPad Prism (Ver.9.0).

## Overexpression of MCL-1

The GFP and MCL-1 expression constructs were obtained from Horizon Discovery. To produce lentiviruses, plasmids were transiently co-transfected into 293T cells with the packaging plasmids pMD2.G, (addgene plasmid #12259) and psPAX2 (addgene plasmid #12260) using Lipofectamine 3000 (Thermo Fisher Scientific). After 48 h, supernatant containing lentiviruses was harvested and filtered with 0.45-μm PVDF filters. Target cells were transduced with the lentiviruses in the presence of 5 mg/mL polybrene for 48 h. After an additional 3-day culture, Blasticidin at 5 μg/ml was used for selection.

## Real-time PCR

RNA extractions were performed using the RNeasy Plus Mini Kit (Qiagen, Germantown, MD) according to the manufacturer's protocol. Complementary DNA (cDNA) was synthesized from RNA by reverse transcription PCR using RNA to cDNA EcoDry™ Premix (TAKARA) and real-time PCR was performed using TaqMan Fast Advanced Master Mix (Invitrogen) according to the manufacturer's protocol. Triplicate PCR reactions were run on ABI 7500 real-time quantitative PCR system. (Applied Biosystems) and $2^{\Delta\Delta}$ method were used for comparative Ct. GAPDH was used as the internal control for these calculations.

## EdU-DAPI-based flow cytometry

$5-10 \times 10^5$ cells were plated in 10 cm dishes and treated with DMSO, 1 nM RMC-6272, 100 nM RM-018 or combination for 24 h. Before harvesting, cells were incubated with 10 μM EdU for 2 h. The cells were harvested and stained with Click-iT EdU Alexa Fluor 488 Flow Cytometry Assay Kit (Invitrogen) according to the manufacturer's protocol. Data were obtained on a Cytek Aurora (Cytek) flow cytometer and analyzed with FlowJo software (version 11.2, BD Biosciences). Gating strategy was done using preliminary SSC-A and FSC-A gates, followed by SSC-H and SSC-W, and then FSC-H and FSC-W. Gating boundaries were determined by negative and positive controls.

## Annexin V-propidium iodide (PI) assay

$5-10 \times 10^5$ cells were plated in 10 cm dishes and treated with DMSO, 1 nM RMC-6272, 100 nM RM-018 or combination. After 72 h, floating cells in media and adherent, trypsinized cells were collected in a single tube and stained with annexin V and propidium iodide using the FITC annexin V Apoptosis Detection Kit I (BD Biosciences) according to the manufacturer's protocol. Data were obtained on a Cytek Aurora (Cytek) flow cytometer and analyzed with FlowJo software (version 11.2, BD Biosciences). Gating strategy was done using preliminary SSC-A and FSC-A gates, followed by SSC-H and SSC-W, and then FSC-H and FSC-W. Gating boundaries were determined by negative and positive controls.

## Xenograft studies

For the CDX experiments, athymic strain #007850 6–10 week-old female mice were used. For the PDX experiments, NSG strain #005557 6–10 week-old female mice were used. Once the mean tumor volume reached ~100–150 mm³, mice were randomized and treated with either vehicle, RMC-4998 (80 mg/kg, once a day, 5 times per week, oral gavage), RMC-6272 (6 mg/kg, once a week, IP), or the combination of these drugs at the same doses as mono-therapies. Similarly, mice were randomized and treated with either RMC-6291 (100 mg/kg once daily, oral gavage), RMC-5552 (10 mg/kg, once a week, IP), or the combination at the same doses as monotherapies. All mice were sacrificed when the maximal tumor volume of 1500 mm³ was reached.

Tumors were measured twice a week using calipers in a non-blinded manner by a research technician who was not aware of the objectives of the study, and their volumes were calculated as width² × length × 0.5. Body weights were monitored twice a week. Animal experiments performed at Memorial Sloan Kettering (MSK) were done according to the protocol approved by the MSK Animal Care and Use Committee. For studies conducted at Wuxi Apptec and Pharmaron, all animal experiments were performed according to the protocol approved by local Animal Care and Use Committee.

## Immunohistochemistry

All tissues were fixed for up to 24 h using 10% neutral buffered formalin and then moved to 70% ethanol for long-term storage. All IHC staining was performed on 4-μm tissue sections using the Leica BOND III autostainer. Primary antibodies were detected with the Leica BOND Polymer Detection kit. Stained slides were scanned and digitized with a 3DHistotech Panoramic whole slide scanner at 40× magnification. Image analysis was performed using HALO software from Indica Labs. Tumor and stromal regions were classified using HALO's Tissue Classifier module and markers quantified only in annotated tumor regions. The following primary antibodies from Cell Signaling Technologies were used, pERK (#4370, 1:1000 dilution), p4E-BP1 (#2855, 1:1500 dilution), p6RP (#5364, 1:400 dilution), Ki67 (#12202S, 1:1000 dilution), and Cleaved Caspase 3 (#9661, 1:400 dilution). Leica poly-HRP was used as the secondary antibody.

## Statistics and reproducibility

Data from RT-PCR, flow cytometry and Caspase 3/7 Activity Assay are presented as mean ± standard deviation (SD), and the tumor progression in animal studies as means ± standard error (SE), respectively. Student's t test (two tailed) or one-way ANOVA, followed by post hoc pairwise analysis test, was performed using GraphPad Prism version 9.00 (GraphPad Software, La Jolla, CA, USA, www.graphpad.com) as indicated in the figure legend. P value < 0.05 was considered significant. The stars in the graphs indicate significance, as detailed in the figure legends. Samples size was determined based on the expected effect size and the statistical power needed to determine significance. No data was excluded from the analyses. The mice experiments were randomized as described. Western blots were repeated in at least two independent experiments.

## Reporting summary

Further information on research design is available in the Nature Portfolio Reporting Summary linked to this article.

# Data availability

All data included in this study is provided in the article, supplementary information, or source data file. Source data are provided with this paper.

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

## Acknowledgements

We thank Zhan Yao for his assistance with experiments and helpful discussions. We thank the Charles Rudin lab for assistance with establishing the PDX models. We thank the Emily Chang lab for providing cell lines. Funding: This research was supported by grants (to N.R.) from the National Institutes of Health (NIH) P01-CA129243; R35 CA210085; the Geoffrey Beene Cancer Research Center; the Emerson Collective Research Grant; Melanoma Research Alliance; The NIH MSKCC Cancer Center Core Grant P30 CA008748 and Experimental Therapeutics Center. P.H.C. is supported by grants from the Druckenmiller Center for Lung Cancer Research, Conquer Cancer the ASCO Foundation, and the Clinical and Translational Science Center at Weill Cornell Medical Center (UL1TR00457).

## Author contributions

H.K., P.H.C., M.S., and N.R. conceived the hypotheses, designed and analyzed the experiments, and wrote the manuscript. D.W. and J.A.M.S. analyzed the experimental data and critically reviewed the manuscript. H.K., P.H.C., Y.C.Y., A.W., P.P., C.T., J.R., J.J., J.E., B.L., K.C., and V.M. established the in vitro and in vivo experimental systems, performed the laboratory experiments, and analyzed the results. E.d.S designed and analyzed the in vivo experiments. J.A.B., H.T., R.Y., and H.E. contributed to the experimental design and data analysis.

## Competing interests

N.R. is on the scientific advisory board (SAB) and owns equity in Beigene, Zai Labs, MapKure, Ribon and Effector. N.R. is also on the SAB of Astra Zeneca and Chugai and a past SAB member of Novartis, Millennium-Takeda, Kura, and Araxes. N.R. is a consultant to RevMed, Tarveda, Array-Pfizer, Boeringher-Ingelheim and Eli Lilly. He receives research funding from Revmed, AstraZeneca, Array Pfizer and Boerhinger-Ingelheim and owns equity in Kura Oncology and Fortress. Y.C.Y., J.J., J.E., B.L., D.W., J.A.M.S., and M.S., are employed by Revolution Medicines. The remaining authors declare no competing interests.
