## [Peer Review File · Nature Communications]

Reviewers' Comments:

Reviewer #1:

Remarks to the Author:

In this report, Kitai and colleagues demonstrate the KRAS(G12C) lung cancer cell lines with varying degrees of intrinsic resistance to KRAS(G12C) inhibitors, including new, potent RAS(G12C) ON inhibitors, exhibit residual mTORC1 activity even in the setting of KRAS inhibition. They demonstrate that mTORC1 inhibition, most notably using bi-steric mTORC1-selective inhibitors, overcomes intrinsic resistance to KRAS blockade, and that combined KRAS and mTORC1 inhibition drives cell cycle arrest (through inhibition of Cyclin D1 expression) and apoptosis (through BIM upregulation and MCL1 suppression), both of which are related to the suppression of cap-dependent translation downstream of mTORC1.

The key results of this study are not particularly surprising given the prior work in the field, including by the Rosen lab, that has demonstrated the importance of mTORC1/4E-BP1 signaling in the survival of RAS/MAPK/PI3K altered tumors treated with their corresponding targeted therapies. However, the results have immediate clinical relevance based on: (1) the strong data in highly relevant in vitro and in vivo models; (2) the availability of exciting, clinical KRAS(G12C)-ON inhibitors and bi-steric mTORC1-selective inhibitors, and (3) the fact that the excellent safety profile of KRAS(G12C) inhibitors, and the potentially tolerable safety profile of bi-steric mTORC1 inhibitors, may enable their safe administration where inhibitors of PI3K/AKT/mTOR, combined with inhibitors of MEK/ERK, previously failed because of toxicities. Finally, the work is notably thorough and carefully done.

This study suggests mechanistic questions that are only partially answered but that, in my mind, are outside of the scope of the current manuscript. These include, as the authors have mentioned, the question of how mTORC1 activity is maintained in the setting of KRAS inhibition. There are also opportunities for further, more resolved dissection of the downstream mechanisms associated with KRAS/mTORC1 inhibitor synergy, but again, these studies are outside of the scope of the current manuscript in my view.

My only request is that the authors consider citing other studies which have similarly made the point that mTORC1 inhibition is essential for the clinical activity of therapies targeting upstream RTK/RAS/RAF/PI3K signaling. These studies include but are not limited to PMID 23903755 and 23903756 (the latter of which features Dr. Rosen as a co-author). An addition to the Introduction or Discussion sections that covers these and other key, related studies would seem to benefit this manuscript without in any way subtracting from its importance.

Reviewer #2:

Remarks to the Author:

The manuscript by Kitai et al. investigates mechanisms that limit the efficacy of KRASG12C inhibitors in NSCLCs and describe a new therapeutic approach of combining KRASG12C and mTORC1 selective inhibitors. The authors begin by showing the enhanced (in vitro) efficacy of two active state KRASG12C inhibitors (tool compounds) and note that these agents more rapidly and more durably suppress ERK signaling. Nevertheless, while more effective, the authors report that the relative activity of these and other RASG12C(off) inhibitors does not completely correlate with robust suppression of the ERK pathway, but instead appears to be more correlated with suppression of PI3K/mTOR pathway components. They then show that selective mTORC1 inhibitors potentially enhance the effects of KRASG12C inhibitors in vitro and report impressive in vivo responses as well in rigorous models.

They further demonstrate that KRASG12C and mTORC1 inhibitors cooperatively suppress cyclin D1 and MCL1, whereas KRASG12C inhibitors are sufficient to induce BIM expression. The importance

of concomitant BIM upregulation and MCL1 suppression in this mediating therapeutic response to these agents is supported by gain and loss of function experiments. The critical role of eIF4E and cap-dependent translation (of MCL1) is also supported by genetic experiments. Finally, the authors show that the induction of apoptosis stimulates a feedforward loop causing further inhibition of cap-dependent translation via the cleavage of eIF4G.

The preclinical efficacy of these agents is extremely impressive and the authors effectively elucidated the mechanism by which they function. While not all aspects of the mechanistic dissection are completely novel, they were essential for this study and the eIF4G degradation piece is interesting and intriguing. Moreover, this does not detract from the importance of this paper or its role in inspiring the development of new clinical trials.

Major:

Line 59. It is more accurate to say "less potent and variable" inhibition of PI3K/mTOR signaling based on the data.

When discussing effects on the PI3K/mTOR pathway, it would be more accurate to say something like "resistant lines exhibit no suppression of AKT, pS6K or 4EBP1 phosphorylation, whereas suppression of all three components is observed in sensitive lines, albeit with different kinetics." This part is important because it demonstrates that KRAG12(on) inhibitors can inhibit the PI3K/AKT segment of the pathway. The authors do touch on this in the discussion, but it is a point of contention. Then they should more accurately describe the variability of the suppression of pAKT, p4EBP1, and pS6K. This would be beneficial to the reader because as the authors know the ERK pathway can feed into this pathway in different ways, which also should be included in the discussion. Incidentally, by eye, efficacy appears to be most closely (dominantly) associated with 4EBP1 phosphorylation (and less so the others).

On that note it I assume the phosphorylation state was calculated based on phosphorylation levels as compared to total protein (of each component individually). That should be more clearly stated in the legend. In addition, the authors should describe what the 23 or 24 samples were to generate the phosphorylation/viability correlation plots- how many lines, duplicates?

The only functional study that is missing is one showing how important Cyclin D1 is for the phenotype. Is it essential for growth arrest (probably), apoptosis (maybe). Any result would likely add to the paper.

Minor:

163 I believe Fig. 2b should be Fig. 2C, Extended data 2a↯ED 2b

The waterfall plots should be below the graphs of tumors from the same animals for clarity.

I believe the LU65 cell line (which is discussed) was omitted from Fig. 1b (right)

Responses to Reviewer Comments

Responses here in red. Changes in the manuscript are highlighted.

Reviewer #1 - Cell death mechanisms, synergy, resistance (xRemarks to the Author):

In this report, Kitai and colleagues demonstrate the KRAS(G12C) lung cancer cell lines with varying degrees of intrinsic resistance to KRAS(G12C) inhibitors, including new, potent RAS(G12C) ON inhibitors, exhibit residual mTORC1 activity even in the setting of KRAS inhibition. They demonstrate that mTORC1 inhibition, most notably using bi-steric mTORC1-selective inhibitors, overcomes intrinsic resistance to KRAS blockade, and that combined KRAS and mTORC1 inhibition drives cell cycle arrest (through inhibition of Cyclin D1 expression) and apoptosis (through BIM upregulation and MCL1 suppression), both of which are related to the suppression of cap-dependent translation downstream of mTORC1.

The key results of this study are not particularly surprising given the prior work in the field, including by the Rosen lab, that has demonstrated the importance of mTORC1/4E-BP1 signaling in the survival of RAS/MAPK/PI3K altered tumors treated with their corresponding targeted therapies. However, the results have immediate clinical relevance based on: (1) the strong data in highly relevant in vitro and in vivo models; (2) the availability of exciting, clinical KRAS(G12C)-ON inhibitors and bi-steric mTORC1-selective inhibitors, and (3) the fact that the excellent safety profile of KRAS(G12C) inhibitors, and the potentially tolerable safety profile of bi-steric mTORC1 inhibitors, may enable their safe administration where inhibitors of PI3K/AKT/mTOR, combined with inhibitors of MEK/ERK, previously failed because of toxicities. Finally, the work is notably thorough and carefully done.

This study suggests mechanistic questions that are only partially answered but that, in my mind, are outside of the scope of the current manuscript. These include, as the authors have mentioned, the question of how mTORC1 activity is maintained in the setting of KRAS inhibition. There are also opportunities for further, more resolved dissection of the downstream mechanisms associated with KRAS/mTORC1 inhibitor synergy, but again, these studies are outside of the scope of the current manuscript in my view.

These are logical questions that stem from this work and we are actively investigating them currently. However, we agree that they are outside the scope of this manuscript.

My only request is that the authors consider citing other studies which have similarly made the point that mTORC1 inhibition is essential for the clinical activity of therapies targeting upstream RTK/RAS/RAF/PI3K signaling. These studies include but are not limited to PMID 23903755 and 23903756 (the latter of which features Dr. Rosen as a co-author). An addition to the Introduction or Discussion sections that covers these and other key, related studies would seem to benefit this manuscript without in any way subtracting from its importance.

Thank you for this important suggestion. We have added several references (listed below) to previous work studying mTORC1 combinations and have also made a change in the discussion to highlight these studies (lines 415-418).

- Elkabets, M., Vora, S., Juric, D., Morse, N., Mino-Kenudson, M., Muranen, T., Tao, J., Campos, A.B., Rodon, J., Ibrahim, Y.H., et al. (2013). mTORC1 inhibition is required for sensitivity to PI3K p110 α inhibitors in PIK3CA-mutant breast cancer. *Sci Transl Med* 5. 10.1126/scitranslmed.3005747.
- Corcoran, R.B., Rothenberg, S.M., Hata, A.N., Faber, A.C., Piris, A., Nazarian, R.M., Brown, R.D., Godfrey, J.T., Winokur, D., Walsh, J., et al. (2013). TORC1 suppression predicts responsiveness to RAF and MEK inhibition in BRAF-mutant melanoma. *Sci Transl Med* 5. 10.1126/scitranslmed.3005753.
- Wang, B., Zhang, W., Zhang, G., Kwong, L., Lu, H., Tan, J., Sadek, N., Xiao, M., Zhang, J., Labrie, M., et al. (2021). Targeting mTOR signaling overcomes acquired resistance to combined

BRAF and MEK inhibition in BRAF-mutant melanoma. *Oncogene* 40, 5590–5599. 10.1038/s41388-021-01911-5.

- Pirazzoli, V., Nebhan, C., Song, X., Wurtz, A., Walther, Z., Cai, G., Zhao, Z., Jia, P., de Stanchina, E., Shapiro, E.M., et al. (2014). Acquired resistance of EGFR-mutant lung Adenocarcinomas to Afatinib plus Cetuximab is associated with activation of mTORC1. *Cell Rep* 7, 999–1008. 10.1016/j.celrep.2014.04.014.

Reviewer #2 - KRASi, lung cancer (Remarks to the Author):

The manuscript by Kitai et al. investigates mechanisms that limit the efficacy of KRASG12C inhibitors in NSCLCs and describe a new therapeutic approach of combining KRASG12C and mTORC1 selective inhibitors. The authors begin by showing the enhanced (in vitro) efficacy of two active state KRASG12C inhibitors (tool compounds) and note that these agents more rapidly and more durably suppress ERK signaling. Nevertheless, while more effective, the authors report that the relative activity of these and other KRASG12C(off) inhibitors does not completely correlate with robust suppression of the ERK pathway, but instead appears to be more correlated with suppression of PI3K/mTOR pathway components. They then show that selective mTORC1 inhibitors potently enhance the effects of KRASG12C inhibitors in vitro and report impressive in vivo responses as well in rigorous models.

They further demonstrate that KRASG12C and mTORC1 inhibitors cooperatively suppress cyclin D1 and MCL1, whereas KRASG12C inhibitors are sufficient to induce BIM expression. The importance of concomitant BIM upregulation and MCL1 suppression in this mediating therapeutic response to these agents is supported by gain and loss of function experiments. The critical role of eIF4E and cap-dependent translation (of MCL1) is also supported by genetic experiments. Finally, the authors show that the induction of apoptosis stimulates a feedforward loop causing further inhibition of cap-dependent translation via the cleavage of eIF4G.

The preclinical efficacy of these agents is extremely impressive and the authors effectively elucidated the mechanism by which they function. While not all aspects of the mechanistic dissection are completely novel, they were essential for this study and the eIF4G degradation piece is interesting and intriguing. Moreover, this does not detract from the importance of this paper or its role in inspiring the development of new clinical trials.

Major:

Line 59. It is more accurate to say “less potent and variable” inhibition of PI3K/mTOR signaling based on the data.

We agree that this more accurately describes the data and have made this change.

When discussing effects on the PI3K/mTOR pathway, it would be more accurate to say something like “resistant lines exhibit no suppression of AKT, pS6K or 4EBP1 phosphorylation, whereas suppression of all three components is observed in sensitive lines, albeit with different kinetics.” This part is important because it demonstrates that KRASG12(on) inhibitors can inhibit the PI3K/AKT segment of the pathway. The authors do touch on this in the discussion, but it is a point of contention.

Thank you for this suggestion and we agree that this is a more accurate way to describe the results. We have made the corresponding changes to the manuscript.

Then they should more accurately describe the variability of the suppression of pAKT, p4EBP1, and pS6K. This would be beneficial to the reader because as the authors know the ERK pathway can feed into this pathway in different ways, which also should be included in the discussion. Incidentally, by eye, efficacy appears to be most closely (dominantly) associated with 4EBP1 phosphorylation (and less so the others).

We agree that a more thorough evaluation of these results is warranted and have made several corresponding changes in the results section. Specifically, we have elaborated on the last point which we agree is the most striking and clear pattern in Fig. 2A, that 4EBP1 phosphorylation is clearly suppressed in the sensitive cell lines while it is poorly suppressed in the intermediate/resistant cell lines. We have also added a point in the discussion about cross-talk between the ERK and PI3K pathways (lines 402-404).

On that note it I assume the phosphorylation state was calculated based on phosphorylation levels as compared to total protein (of each component individually). That should be more clearly stated in the legend. In addition, the authors should describe what the 23 or 24 samples were to generate the phosphorylation/viability correlation plots- how many lines, duplicates?

The legend has been changed to better describe the methods. In brief, Fig. 2b was generated with 12 cell lines each plotted twice with the results from two different inhibitors, AZD8037 and Sotorasib. Fig. 2c was generated with 12 cell lines plotted once with results from RM-018.

The only functional study that is missing is one showing how important Cyclin D1 is for the phenotype. Is it essential for growth arrest (probably), apoptosis (maybe). Any result would likely add to the paper.

We thank the reviewer for this important comment and have now attempted to rectify this omission. We have performed several experiments involving siRNA knockdown of Cyclin D1 to evaluate its function in the cell line H1373 (Extended Data Fig. 4c-e, Extended Data Fig. 5c). In sum, these experiments show that knockdown of Cyclin D1 is sufficient to cause cell cycle arrest on par or even greater than the drug combination of RM-018 and RMC-6272, but does not induce apoptosis. These results suggest that a large part of the effect of the drug combination is due to its effect on Cyclin D1. These results offer new questions about how cell cycle and apoptosis regulation affect each other, and this is an area that we are actively investigating, but we believe is beyond the scope of this present manuscript.

Minor:

163 I believe Fig. 2b should be Fig. 2C, Extended data 2a, 2b?

Thank you for this correction, this is now correctly labeled.

The waterfall plots should be below the graphs of tumors from the same animals for clarity.

This has been changed.

I believe the LU65 cell line (which is discussed) was omitted from Fig. 1b (right)

Thank you for catching this error. This has been corrected.

Reviewers' Comments:

Reviewer #1:

Remarks to the Author:

The authors have satisfactorily addressed all of my requests. Congratulations on a very nice and important study.

Reviewer #2:

Remarks to the Author:

I am satisfied with the revisions